# A Scoring System to Predict Severe Acute Lower Respiratory Infection in Children Caused by Respiratory Syncytial Virus

**DOI:** 10.3390/microorganisms12071411

**Published:** 2024-07-12

**Authors:** Ri De, Mingli Jiang, Yu Sun, Siyuan Huang, Runan Zhu, Qi Guo, Yutong Zhou, Dong Qu, Ling Cao, Fengmin Lu, Linqing Zhao

**Affiliations:** 1Department of Microbiology and Infectious Disease Center, School of Basic Medical Sciences, Peking University Health Science Center, Beijing 100191, China; graceride@163.com; 2Laboratory of Virology, Beijing Key Laboratory of Etiology of Viral Disease in Children, Capital Institute of Pediatrics, Beijing 100020, China; jiangmingli00@163.com (M.J.); sunyu780312@163.com (Y.S.); runanzhu@163.com (R.Z.); g7siete1220@163.com (Q.G.); 18601399785@163.com (Y.Z.); 3Department of Intensive Care Unit, Affiliated Children’s Hospital, Capital Institute of Pediatrics, Beijing 100020, China; hsy1312@126.com (S.H.); qudong2012@126.com (D.Q.); 4Department of Respiratory Medicine, Affiliated Children’s Hospital, Capital Institute of Pediatrics, Beijing 100020, China; caoling9919@163.com

**Keywords:** human respiratory syncytial virus, severe acute lower respiratory tract infection, risk factors, scoring system, children

## Abstract

There were several factors associated with respiratory syncytial virus (RSV) severe acute lower respiratory infection (RSV-sALRI) in infants and young children. It is vital to develop a convenient scoring system to predict RSV-sALRI in children. Pediatric patients with RSV-ALRI from January 2009 to December 2021 were recruited retrospectively. Two-third of them were randomly grouped into the development set and one-third to the validation set. In the development set, risk factors for RSV-sALRI were transferred into the logistic regression analysis, then their receiver operating characteristic (ROC) curves were built to obtain the area under the ROC curve (AUC), and regression coefficients for each predictor were converted to points. Finally, the value of the scoring system was evaluated in the validation set. A total of 1 066 children with RSV-ALRI were recruited, including 710 in the development set and 356 in the validation set. By logistic regression analysis, six factors (younger than 2 years, gestational age <37 weeks, have siblings, birth weight ≤2500 g, artificial/mix feeding, CHD) showed statistical difference and then were scored with points according to the coefficient value (OR) in the development set. In the validation set, the sensitivity of the scoring system was 70.25%, the specificity 85.53%, the positive predictive value 71.43%, the negative predictive value 84.81%, and coincidence rate 0.80. The Kolmogorov–Smirnov test showed the distribution of AUC 0.765 (SE = 0.027; 95% CI = 0.713–0.818; *p* < 0.001). A simplified scoring system was developed in the study with high prediction value for RSV-sALRI in children.

## 1. Introduction

Acute lower respiratory tract infection (ALRI) is a major threat to global public health, mainly in middle-income countries, resulting in a high disease burden every year due to death or disability [1,2]. Human respiratory syncytial virus (RSV) is the most common viral pathogen and the main cause of ALRI, leading to a large number of hospitalizations and high death burden in infants and young children [3,4,5]. It is globally estimated that RSV is associated with about 22% of all ALRI, and there were 3.6 million hospital admissions for RSV-ALRTI and 26,300 in-hospital RSV-ALRTI deaths in 2019 [4]. In children younger than 5 years, more than 33 million new episodes of ALRI per year were RSV positive and about 10% of these were severe enough to warrant hospitalization with around 200,000 in ICU and in-hospital deaths one year globally [6,7,8]. In children aged 0–6 months, there were 1.4 million RSV-ALRTI hospital admissions, resulting in a total of 13,300 deaths worldwide in 2019 [4]. The vast majority of children have had serologically proven infection with RSV by age 2 years, representing a major health-care burden [9,10]. The global cost of RSV-sALRI management of pediatric patients in 2017 was approximately EUR 4.82 billion [11]. The estimated median direct cost of RSV-associated hospitalization in children under five years was USD 10 million [12].

According to the systematic reviews and meta-analysis, it has been reported that there were several factors associated with RSV-sALRI in young children (younger than 2 years) such as prematurity, low birth weight, being male, having siblings, maternal smoking, history of atopy, no breastfeeding, crowding, and congenital heart disease [13,14]. It also has been reported that there are two major RSV subtypes, A and B, and multiple genotypes, which can coexist during the RSV epidemic season every year and result in different disease severity. Compared to those infected by RSV-B, more patients infected by RSV-A required mechanical ventilation (*p* = 0.01) and intensive care (*p* = 0.008) [15].

With the approval of two pre-F-based RSV vaccines, Arexvy^®^ (GSK, London, UK) and Abrysvo^®^ (Pfizer, New York, NY, USA), offering good protection against RSV infection in older people and to infants by immunizing pregnant individuals and a long-acting humanized monoclonal antibody Nirsevimab-alip (Beyfortus^®^, AstraZeneca, Cambridge, UK), marking a breakthrough in the prevention of RSV in infants and young children by the United States Food and Drug Administration (FDA) [16,17]. The year 2023 marks a milestone for RSV prevention. All these prevention managements try to alleviate the incidence of RSV-sALRI. Therefore, a scoring system to grade the effects of risk factors on RSV-sALRI would significantly contribute to the targeting prevention of RSV-sALRI.

Therefore, in order to develop a convenient scoring system, the clinical information of pediatric patients aged 0–16 years with RSV-associated ALRI from January 2009 to December 2021 were collected retrospectively in this study, and a simplified score system with prediction points ranging 0–11 associated with risk factors was established, which was expected to help clinicians in preliminary assessing the risk of severe disease and choosing an appropriate treatment and timely intervention measures.

## 2. Materials and Methods

### 2.1. Study Design and the Eligible Criteria for Participants

This was a retrospective study conducted in Children’s Hospital of Capital Institute of Pediatrics. Pediatric patients were included in the study according to the eligible criteria: ① aged 0–16 years; ② diagnosed with ALRI; ③ admitted to the Department of Intensive Care Unit (ICU) or Department of Respiratory Medicine from January 2009 to December 2021; and ④ positive for RSV determined by capillary electrophoresis-based multiplex polymerase chain reaction (PCR) (CEMP) assay (Ningbo HEALTH Gene Technologies Ltd., Ningbo, China) and subtyped to A or B [18]. Then, their clinical data, including gender, age, birth weight, gestational age, feeding styles, having or no siblings, symptoms and signs of ALRI, the blood oxygen saturation, and having or not having congenital heart disease (CHD), were collected.

For RSV screening using CEMP assay, total nucleic acid (DNA and RNA) was extracted from 140 µL supernatant of each collected specimen using the QIAamp MinElute Virus Spin Kit (Qiagen GmbH, Hilden, Germany) according to the manufacturer’s instructions. Then, under the direction of the manufacturer’s instructions of the CEMP assay kit for multiplex PCRs, 15 pairs of primers for detecting 13 pathogens, deoxynucleoside triphosphates (dNTPs), MgCl2, and buffer were included. Nucleic acid extracts from clinical specimens were amplified and then subjected to capillary electrophoresis on a GeXP capillary electrophoresis system (Sciex, Concord, ON, Canada). Signals of the 15 labeled PCR products were measured by fluorescence and separated by size: influenza virus (Flu) A 105 nt (2009H1N1 163.3 nt, H3N2 244.9 nt), Flu B 212.7 nt, Adenovirus (AdV) 110.2/113.9 nt (representing different subtypes), human bocavirus (HBoV) 121.6 nt, human rhinovirus 129.6 nt, human parainfluenza virus (PIV) 181.6 nt, chlamydia (Ch) 190.5 nt, human metapneumovirus 202.8 nt, *Mycoplasma pneumoniae* (Mp) 217 nt, human coronavirus (HCoV) 265.1 nt, and RSV 280.3 nt. For RSV subtyping, cDNA obtained by reverse transcription was used as the PCR template. Forward (P4: 5′-TGGGACACTCTTAATCAT-3′) and reverse primers (P5: 5′-TGATTCCAAGCTG AGGAT3′, P6: 5′-GTTGTATGGTGTGTTTC-3′) were used [18].

According to the World Health Organization (WHO), the severity of ALRI is classified into three grades: ① mild ALRI, with acute lower respiratory tract infection symptoms + not requiring hospitalization; ② moderate ALRI, with acute lower respiratory tract infection symptoms + requiring hospitalization and blood oxygen saturation ≥93%; and ③ severe ALRI, with acute lower respiratory tract infection symptoms + requiring hospitalization and blood oxygen saturation <93% [19]. In this study, only hospitalized patients with moderate and severe ALRI were included.

According to clinical data and RSV subtyping results, all cases included in the study were classified into groups of male or female, <2 years or ≥2 years, with birth weight <2.5 kg or ≥2.5 kg, with gestational age <37 weeks or ≥37 weeks, in breast or artificial/mix feeding, having (Yes) or not having (No) siblings, and having (Yes) or not having (No) CHD, with RSV subtyping into A or B.

### 2.2. The Development of the Prediction Model and Scoring System

All cases included in the study were randomly divided into two groups, two-thirds of which were classified as “the development set” to develop the prediction model and scoring system, and one-third as “the validation set” to evaluate the prediction effects of the scoring system.

For cases in the development set, the relationship between RSV-associated sALRI cases and each individual predictor risk factor was evaluated by logistic regression analysis with backwards stepwise selection (*p =* 0.05). Then, all selected predictor variables were used to develop a preliminary risk factor model. The contribution of each risk factor within the model was presented as beta coefficients and odds ratios (ORs) and the adjusted risk of *p* values were two-sided with <0.05 as statistically significant. The OR value was calculated by dividing the ratio of positive to negative numbers in the case group (severe RSV-associated ALRI group) from the ratio of positive to negative numbers in the control group (moderate RSV-associated ALRI group).

Then, the model was verified and modified by 2 methods to find the combined effects of risk factors in keeping the balance between predictive accuracy and simplicity: ① sequential removing and reinsertion of each risk factor variable from the dataset to evaluate its impact on predicting RSV-sALRI and ② using the receiver operating characteristic (ROC) curve by plotting sensitivity against 1-specificity to evaluate the discrimination role of the model to correctly distinguish non-events and events. An area under the ROC curve (AUC) of 0.5 indicates “no” discrimination role, whereas AUC of ≥0.75 indicates “good” discrimination role [20].

The best performance model was used to construct the scoring system, in which rounding-off method (decimals to round up and round down number) was chosen to convert the ORs into score points. Then, the cut-off value of the score system was calculated from the ROC curve and the total score summing up all points of risk factors. Those with score over the cut-off value predicate high possibility of severe RSV-ALRI, whilst those with score under the cut-off value predicate high possibility of moderate RSV-ALRI.

### 2.3. The Verification of the Scoring System

In order to verify the scoring system established by the development set, all risk factors of each registered patient in the verification set were scored to obtain the total score. Those with score over the cut-off value predicate high possibility of severe RSV-ALRI, whilst those with score under the cut-off value predicate high possibility of moderate RSV-ALRI. Then, the sensitivity, specificity, false negative rate, false positive rate, positive predictive value, negative predictive value, and coincidence rate of the scoring system were obtained by comparing with WHO’s severity classification standard, so as to determine the accuracy and effectiveness of the scoring system [21,22]. The internal validity of the scoring system was assessed using 1000 bias-corrected bootstrap samples. Then, a ROC curve was constructed for each case and the dispersion statistics (standard deviation and range) across 100 cases were assessed. A low level of dispersion indicates an internally consistent model. All data analyzed and the ROC curves generated were on the basis of the multivariate prediction model using SPSS 25.0.

## 3. Results

### 3.1. General Information of Participants

From January 2009 to December 2021, there were 1115 pediatric patients diagnosed with ALRI, admitted to the Department of Intensive Care Unit (ICU) or Department of Respiratory Medicine, and positive for RSV. Among these patients, there were 49 excluded from the study, including 15 lacking feeding-related data, 10 with no information of gestational age, and 24 with no birth weight. Therefore, there were 1 066 cases enrolled in the retrospective study.

Among these 1 066 cases, there were 684 (64.17%, 684/1066) male, 827 (77.58%, 827/1066) younger than 2 years (<2 y), 169 (15.85%, 169/1066) with gestational age <37 weeks, 123 (11.54%, 123/1066) in birth weight <2.5 kg, 471 (44.18%, 471/1066) with artificial/mix feeding, 486 having siblings (45.59%, 486/1066), and 115 with CHD (10.79%, 115/1066), while there were 609 (609/1066, 57.13%) infected with subtype A and 457 (457/1066, 42.87%) with subtype B of RSV.

Then, these 1066 cases were assigned to two groups using the random sampling method, including 710 to the development set and 356 to the validation set.

### 3.2. Factors Extracted from the Development Set with Predictive Value for RSV-sALRI

In the development set containing 710 cases, there were 547 (77.04%, 547/710) <2 years, 451 (63.52%, 451/710) male, 105 (14.79%, 105/710) with gestational age <37 weeks, 77 (10.85%, 77/710) with birth weight < 2.5 kg, 306 (43.10%, 306/710) with artificial/mix feeding, 323 having siblings (45.49%, 323/710), and 81 (11.41%, 81/710) with CHD, while 403 (56.76%, 403/710) were infected with subtype A of RSV and 307 (43.24%, 307/710) with subtype B. According to the severity grade standard of the WHO, 477 (67.18%, 477/710) cases were classified into moderate RSV-ALRI and 233 (32.82%, 233/710) as severe RSV-ALRI.

The adjusted risk factors which showed positive and significant associations with RSV-sALRI were age <2 years (OR = 2.81, *p* < 0.001, 95% CI = 1.82–4.32), gestational age <37 weeks (OR = 2.35, *p* < 0.001, 95% CI = 1.37–3.83), having siblings (OR = 2.03, *p* < 0.001, 95% CI = 1.48–2.78), birth weight <2.5 kg (OR = 2.31, *p* <0.001, 95% CI = 1.43–3.72), artificial/mix feeding (OR = 1.42, *p* = 0.028, 95% CI = 1.04–1.93), and having CHD (OR = 2.18, *p* < 0.001, 95% CI = 1.37–3.48). Because there was no association between “male” (OR = 1.19, *p* = 0.903, 95% CI = 0.86–1.64) and “subtype A of RSV” with severe RSV-ALRI in the multivariate analysis (*p* > 0.05), “male” and “subtype A of RSV” were eliminated from risk factors for RSV-sALRI (Table 1).

### 3.3. The Scoring System Developed from the Development Set

In order to easily assess of the risk of RSV-sALRI in pediatric patients, a scoring system was developed by assigning each risk factor with some points according to the OR value in the development set (Table 1). Each OR value was rounded to the nearest whole number for the easy-use scoring system. For example, 3 points were set to pediatric patients “<2 years”, a factor with an OR value of 2.81, while 2 points were set to those with gestational age “<37 weeks”, a factor with an OR value of 2.35, 1 point was set to those with “artificial/mix feeding” style, a factor with an OR value of 1.42, 2 points were set to those with birth weight “<2.5 kg”, a factor with an OR value of 2.31, 2 points were set to those “with sibling”, a factor with an OR value 2.03, and 2 points were set to those with “CHD”, a factor with an OR value of 2.18. The highest point was set to those “<2 years”, the most predictive risk factor for RSV-sALRI.

According to the scoring system that was developed, the highest score of a pediatric patient was 11 points. By the verification of coincidence rate, a score of 5 is the cut-off value, while a score of <5 represents a high possibility of moderate RSV-ALRI, and the score of ≥5 indicates that the possibility of severe RSV-ALRI is high, with the possibility increasing with a higher score (Figure 1).

### 3.4. Internal Validation of the Scoring System Evaluated in the Validation Set

In the validation set containing 356 cases, there were 279 (78.37%, 279/356) <2 years, 38 (10.67%, 38/356) with gestational age <37 weeks, 41 (11.52%, 41/356) with birth weight <2.5 kg, 162 (45.51%, 162/356) with artificial/mix feeding, 194 having siblings (54.49%, 194/356), and 14 (3.93%, 14/356) with CHD.

The risk of RSV-sALRI for the population in the validation set was evaluated using the RSV-sALRI scoring system developed in the study. There were 237 cases (66.57%, 237/356) with points <5, and the median points with quartile were 3 [2,4]. There were 119 cases (33.43%, 119/356) with points ≥5 and the median points with quartile were 6 [5,7].

Then, the disease severity of these pediatric patients in the validation set was evaluated by the severity grade standard of the WHO on the basis of clinical data. There were 235 cases (66.01%, 235/356) classified into moderate RSV-ALRI in which 201 (85.53%, 201/235) were with points <5, predicating high possibility of moderate RSV-ALRI, and 121 into severe RSV-ALRI (33.99%, 121/356) in which 85 (70.25%, 85/121) were with points ≥5, predicating high possibility of severe RSV-ALRI. The median points with quartile were 3 [2,4] in the moderate RSV-ALRI group and 6 [5,8] in the severe RSV-ALRI group, which showed significant differences between the moderate and severe RSV-ALRI group (Z = 13.421, *p* < 0.001).

Compared to the severity grade standard of the WHO, the sensitivity of the scoring system in predicting the severity of RSV-ALRI was 70.25% (85/121), while the specificity was 85.53% (201/235), the false negative rate was 29.75% (36/121), and the false positive rate was 14.47% (34/235). The positive predictive value of the scoring system was 71.43% (85/119), while the negative predictive value was 84.81% (201/237), with coincidence rate 0.80. The area under the ROC of the scoring system was determined by the Kolmogorov–Smirnov test was 0.765 (SE = 0.027; 95% CI = 0.713–0.818; *p* < 0.001) (Figure 2).

## 4. Discussion

The high incidence of RSV infection and its potential severe outcome make it important to identify the risk factors of RSV-sALRI in children. To date, there has been few scoring system to evaluate the association between various risk factors and RSV-severe ALRI in pediatric patients in China. In this study, we aimed to establish a simplified score system in order to help clinicians to assess the risk of severe RSV-associated ALRI and treat it in a more timely manner.

Based on the clinical data from a cohort collected retrospectively over 12 years in Beijing, a simplified scoring system was developed to predict the risk of RSV-sALRI in children aged 0–16 years. Among the eight factors evaluated, there were six factors independently associated with high risk of RSV-sALRI determined by logistic regression analysis, namely, younger than 2 years, with gestational age <37 weeks, having siblings, with birth weight ≤2500 g, in artificial/mix feeding, and having CHD. However, gender and the subtype of RSV were not defined as risk factors for RSV-sALRI in the study compared between groups with moderate RSV-ALRI and severe RSV-ALRI. Then, those six factors associated with high risk of RSV-sALRI were combined by assigning each factor with some points according to its OR value. Finally, a simplified scoring system was built. Compared to the severity grade standard of the WHO, the scoring system showed the sensitivity of 70.25%, the specificity of 85.53%, the false negative rate of 29.75%, the false positive rate of 14.47%, the positive predictive value of 71.43%, and the negative predictive value of 84.81%, with coincidence rate 0.80, while its area under the ROC was 0.765 (SE = 0.027; 95% CI = 0.713–0.818; *p* < 0.001).

In the scoring system, “younger than 2 years” was the strongest predictor of RSV-sALRI in the study with OR value of 2.806 in the logistic regression and scored 3 points in the scoring system, which was supported strongly by previous studies [23,24]. It can be explained by the immature immune system and narrow airways in those younger than 2 years, and a Th2 (T helper 2 cell)-biased immune response caused by RSV infection [25].

“Being male” was not considered a risk factor (OR = 1.187, *p* = 0.343, 95% CI = 0.857–1.643) for RSV-sALRI in the study. However, it has been proved that more cases were male among pediatric patients with RSV-associated pneumonia (male vs. female, 1.425:1) in a study with data accumulated over 30 years [7]. Therefore, “being male” is the undisputed factor for RSV-ALRI. In the study, only patients admitted to the Department of ICU and Department of Respiratory Medicine were enrolled, excluding patients with mild ALRI. No significant difference was shown in “being male” between patients with moderate and severe ALRI.

In the study, CHD was regarded as a high risk factor (OR = 2.183, *p* < 0.001, 95% CI = 1.369–3.481) for RSV-sALRI with an OR value of 2.183 and 2 points in the scoring system. It has been reported that infants and young children with CHD were more often admitted to ICU for severe ALRI when they were infected by RSV, with supplemental oxygen therapy or prolonged mechanical ventilation, and the mortality of them increased 2–6-fold compared with children without any risk factors of RSV-sALRI, The occurrence of CHD in infants and young children seriously limits the cardiac output and oxygen delivery. When infants and young children are infected with RSV, the situation will be worse [26].

The role of breastfeeding in preventing RSV-ALRI is indisputable, because RSV-IgA and lactoferrin in breast milk have protective effects and may promote maturity through the influence of prolactin [27]. Contrary to the protective effect of breastfeeding, artificial/mixed feeding was a risk factor of RSV-sALRI in this study (OR = 1.417, *p* = 0.028, 95% CI = 1.037–1.937).

Another strong risk factor for RSV-sALRI reported in previous studies was low birth weight (≤2500 g), which was included in the logistic model (OR = 2.307, *p* < 0.001, 95% CI = 1.431–3.718) in the study, as well as low gestational age <37 weeks, which showed a OR value of 2.349 in the logistic regression analysis (OR = 2.349, *p* < 0.001, 95% CI = 1.366–3.832). These can be explained by the large proportion of infants with gestational age <37 weeks that had low birth weight [28,29]. As reported in some studies [30], young children having siblings were at high risk for RSV-sALRI. “Having siblings” was also associated with high risk of RSV-sALRI (OR = 2.030, *p* < 0.001, 95% CI = 1.482–2.781) in the study.

However, there are some limitations in the study. Firstly, many risk factors of RSV-sALRI reported in previous studies [31,32], such as race/ethnicity, parents’ educational level or social conditions, allergy history of family, and exposure to cigarette smoking, were not evaluated in the study. There are several reasons. The first one is that only one race/ethnicity population were enrolled in the study. No clinical data were collected concerning the information about parents’ educational level or social conditions. Only partial clinical data were collected about the allergy history of family and exposure to cigarette smoking. In order to develop a more accurate scoring system, more risk factors should be assessed by accumulating more detailed clinical data. Secondly, this was a single-center study, with a total of 1 066 cases and a scoring system verified by internal set. Particularly, there were some cases that were enrolled during the outbreak of severe acute respiratory syndrome coronavirus 2 (SARS-CoV-2) under the pressure of nonpharmaceutical interventions for coronavirus disease (COVID-19) in Beijing, China, and the endemic patterns of RSV infection in pediatric patients has changed. Therefore, more data should be accumulated to evaluate the value of the scoring system.

In conclusion, a simplified scoring system with points ranging 0–11 was developed, and has been confirmed with high prediction value in predicting the risk of severe RSV-ALRI in children aged 0–16 years. Moreover, some prospective research work in multi-centers should be carried out to test its applicability to clinical practice.

## Figures and Tables

**Figure 1 microorganisms-12-01411-f001:**
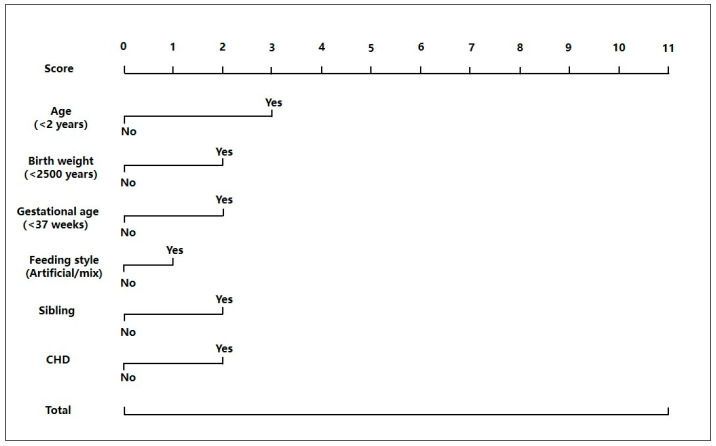
Score values assigned for each risk factor in scoring system to predict the risk of RSV-sALRI.

**Figure 2 microorganisms-12-01411-f002:**
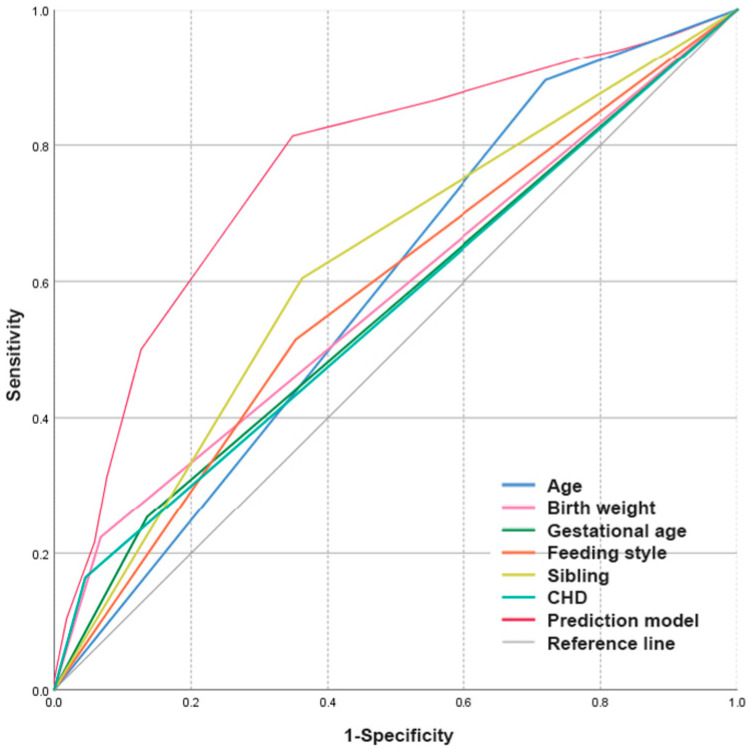
The receiver-operating characteristic (ROC) curves of different points assigned for the risk factors and the prediction model of the scoring system developed by combining risk factors together on predicting RSV-sALRI were shown to count their area under the ROC curve (AUC). The AUC of the scoring system is 0.765 (SE = 0.027; 95% CI = 0.713–0.818; *p* < 0.001).

**Table 1 microorganisms-12-01411-t001:** Risk factors evaluated by comparison between pediatric patients with severe and moderate RSV-ALRI in the development set.

Factors		No. Enrolled (%)	Severe ALRI No. (%)	Moderate ALRI No. (%)	χ^2^, *p* Value	OR Value	95% CI
Gender	Male	451	160	291	0.903, 0.343	1.19	0.86–1.64
	Female	259	82	177			
Age	<2 year	547	212	335	**23.152, 0.000**	**2.81**	**1.82–4.32**
	≥2 year	163	30	133			
Gestational age	<37 week	105	54	51	**15.605, 0.000**	**2.35**	**1.37–3.83**
	≥37 week	605	188	417			
Birth weight	<2500 g	77	40	37	**12.267, 0.000**	**2.31**	**1.43–3.72**
	≥2500 g	633	202	431			
Feeding style	Breast feeding	404	124	280	**4.799, 0.028**	**1.42**	**1.04–1.94**
	artificial/mix	306	118	188			
With siblings	Yes	323	138	185	**19.689, 0.000**	**2.03**	**1.48–2.78**
	No	387	104	283			
With CHD	Yes	81	41	40	**11.123, 0.001**	**2.18**	**1.37–3.48**
	No	629	201	428			
Subtype of RSV	RSV A	405	126	279	3.710, 0.054	0.736	0.54–1.01
	RSV B	305	116	189			

Note: boldfaced character is *p* < 0.05, significant difference observed.

## Data Availability

The original contributions presented in the study are included in the article, further inquiries can be directed to the corresponding authors.

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
