# Peer review of "A Scoring System to Predict Severe Acute Lower Respiratory Infection in Children Caused by Respiratory Syncytial Virus"

_microorganisms, 2024, doi:10.3390/microorganisms12071411_

Round 1
Reviewer 1 Report
Comments and Suggestions for Authors
Comments and suggestions
Title section:
1. Title: specify the age range of the children Summary section: 2. Can patients be recruited in a retrospective study? or the necessary information was obtained from information sources. 3. Place the ranges of the scoring system.
Introduction section: 4. Enter the age ranges of the babies and children mentioned in this section. 5. Quantifies serious illness, ICU admission and increase in days of hospitalization. 6. Add information on the burden of disease caused by RSV. 7. Explicitly state the objective of the study 8. The last introductory paragraph is describing part of the methodology so it must be restructured.
Materials and methods section: 9. It is not clear how the necessary data to carry out this retrospective was obtained. The authors state "patients were recruited" but recruiting is typical of a prospective study. In this case, were clinical records reviewed that met the inclusion criteria/definitions? 10. Explain in a paragraph how RSV infection and severe respiratory tract infection were defined. 11. Was a control group sought with healthy individuals or those infected by other microorganisms? 12. Restructure the scoring system verification paragraph for better understanding. Results Section: 13. This section would benefit from a table containing data from the case studies. Discussion Section: 14. Restructure paragraphs 1 and two based on the objective of the study. Conclusions Section: 15. Restructure conclusions based on the objectives of the study
Comments on the Quality of English LanguageMinor editing of English language required
Author Response
Comments 1: [Title section: 1. Title: specify the age range of the children]
Response 1: Thank you for pointing this out. We agree with this comment. Therefore, we revised the title to “A scoring system to predict severe acute lower respiratory infection in children aged 0-16 years caused by respiratory syncytial virus”. This change can be found in Title page.
Comments 2: [Summary section: 2. Can patients be recruited in a retrospective study? or the necessary information was obtained from information sources. ]
Response 2: Thank you for pointing this out. We agree with this comment. We revised the sentence to “Clinical information of pediatric patients aged from 0-16 years with RSV-associated ALRI from Jan 2009 to December 2021 were collected retrospectively.” in method section of Abstract.
Comments 3: [ Summary section: 3. Place the ranges of the scoring system.]
Response 3: Thank you for pointing this out. We agree with this comment. We revised the sentence to “A simplified scoring system with points ranging 0-11 was developed in the study with high prediction value for RSV-sALRI in children.” in summary section of Abstract.
Comments 4: [Introduction section: 4. Enter the age ranges of the babies and children mentioned in this section.]
Response 4: Thank you for pointing this out. We agree with this comment. In page 2, line 58 in Introduction, the revised version is “In this study, the clinical information of pediatric patients aged 0-16 years with RSV-associated ALRI from Jan 2009 to December 2021 were collected retrospectively.”
Comments 5 [ Introduction section: 5. Quantifies serious illness, ICU admission and increase in days of hospitalization.]
Response 5: Thank you for pointing this out. We agree with this comment. The revised version is “It is globally estimated that RSV is associated with about 22% of all ALRI, and there were 3.6 million hospital admissions for RSV-ALRTI, and 26,300 in-hospital RSV-ALRTI deaths in 2019 [4]. In children younger than 5 years, more than 33 million new episodes of ALRI per year were RSV positive and about 10% of these were severe enough to warrant hospitalization with around two hundred thousand in ICU and in-hospital deaths one year globally [6-8]. In children aged 0-6 months, there were 1.4 million RSV-ALRTI hospital admissions, resulting in a total of 13,300 deaths worldwide in 2019 [4].” This change can be found page 1, lines 6-12.
Comment 6 [Introduction section: 6. Add information on the burden of disease caused by RSV. ]
Response 6: Agree. We added the burden of disease caused by RSV in Introduction section. “The global cost of RSV-sALRI management of pediatric patients in 2017 was approximately €4.82 billion [11]. And the estimated median direct cost of RSV-associated hospitalization in children under five years was $10 million [12].” This change can be found in page 1, lines 13-15.
Comment 7 [Introduction section:7. Explicitly state the objective of the study.]
Response 7: Thank you for pointing this out. We state the objective of the study as “In order to develop a convenient scoring system.” in page 2, line 171.
Comment 8 [Introduction section:8. The last introductory paragraph is describing part of the methodology so it must be restructured.
Response 8: Thank you for pointing this out. We revised the last introductory paragraph to “In order to develop a convenient scoring system, the clinical information of pediatric patients aged 0-16 years with RSV-associated ALRI from Jan 2009 to December 2021 were collected retrospectively in this study, and a simplified score system with prediction points ranging 0-11 associated with risk factors was established, which was expected to help clinicians in preliminary assessing the risk of severe disease and choosing an appropriate treatment and timely intervention measures.”. This change can be found in page 2, lines 171-175.
Comment 9 [Materials and methods section: 9. It is not clear how the necessary data to carry out this retrospective was obtained. The authors state "patients were recruited" but recruiting is typical of a prospective study. In this case, were clinical records reviewed that met the inclusion criteria/definitions? ]
Response 9: Thank you for pointing this out. The revised version is “This was a retrospective study conducted in Children’s Hospital of Capital Institute of Pediatrics. Pediatric patients were included in the study according to the eligible criteria:â‘ aged 0-16 years; â‘¡diagnosed with ALRI; â‘¢ admitted to the Department of Intensive Care Unit (ICU) or Department of Respiratory Medicine during January 2009 to December 2021; â‘£ positive for RSV determined by capillary electrophoresis-based multiplex polymerase chain reaction (PCR) (CEMP) assay (Ningbo HEALTH Gene Technologies Ltd., Ningbo, China) [17] and subtyped to A or B [18]. Then, their clinical data, including gender, age, birth weight, gestational age, feeding styles, having or no siblings, symptoms and signs of ALRI, the blood oxygen saturation, and having or no congenital heart disease (CHD) were collected.” in page 2, lines 179-187.
Comment 10 [Materials and methods section: 10. Explain in a paragraph how RSV infection and severe respiratory tract infection were defined. ]
Response 10: Thank you for pointing this out. We defined the severity of ALRI as “According to the World Health Organization (WHO), the severity of ALRI is classified into three grades: â‘ mild-ALRI, with acute lower respiratory tract infection symptoms +, not requiring hospitalization; â‘¡ moderate-ALRI, with acute lower respiratory tract infection symptoms +, requiring hospitalization, blood oxygen saturation ≥93%; â‘¢ severe-ALRI, with acute lower respiratory tract infection symptoms +, requiring hospitalization, and blood oxygen saturation <93% [19]. In this study, only hospitalized patients with moderate- and severe-ALRI were included.” This change can be found in page 2, lines 188-193.
Comment 11 [Materials and methods section: 11. Was a control group sought with healthy individuals or those infected by other microorganisms? ]
Response 11: Thank you for pointing this out. However, no control group with healthy individuals or those infected by other microorganisms were sought in the study. The aim of the study is to reveal the effect of different risk factors on the severity RSV-ALRI. Therefore, we paid more attention to risk factors, instead of healthy individuals or those infected by other microorganisms.
Comment 12 [Materials and methods section: 12. Restructure the scoring system verification paragraph for better understanding.
Response 12: Thank you for pointing this out. We have restructured the scoring system verification paragraph for better understanding. “In order to verify the scoring system established by the development set, all risk factors of each registered patient in the verification set were scored to get the total score. Those with score over the cut-off value predicate high possibility of severe RSV-ALRI, whilst those with score under the cut-off value predicate high possibility of moderate RSV-ALRI. Then, the sensitivity, specificity, false negative rate, false positive rate, positive predictive value, negative predictive value, and coincidence rate of the scoring system are obtained by comparing with WHO's severity classification standard, so as to determine the accuracy and effectiveness of the scoring system [21, 22].” This change can be found in page 4, lines 494-500 of Materials and methods section.
Comment 13 [Results Section: This section would benefit from a table containing data from the case studies.
Response 13: Thank you for pointing this out. The data of risk factors were shown in Table 1 which has been added in page 5, line 574.
Comment 14 [Discussion Section: Restructure paragraphs 1 and two based on the objective of the study.]
Response 14: Thank you for pointing this out. We restructured paragraphs 1 and 2 based on the objective of the study. “The high incidence of RSV infection and its potential severe outcome make it important to identify the risk factors of RSV-sALRI in children. To date, there has been few scoring system to evaluate the association between various risk factors and RSV-severe ALRI in pediatric patients in China. In this study, we aimed to establish a simplified score system in order to help clinicians to assess the risk of severe RSV-associated ALRI and treat more timely.
Based on the clinical data from a cohort collected retrospectively over 12 years in Beijing,, a simplified scoring system was developed to predict the risk of RSV-sALRI in children aged from 0-16 years. There were six factors independently associated with high risk of RSV-sALRI determined by logistic regression analysis, namely, younger than 2 years, with gestational age < 37 weeks, having sibling, with birth weight ≤2,500g,in artificial/mix feeding, and with CHD. Then, these six factors were combined together by assigning each factor with some points according to its OR value, and a scoring system was built. Compared to the severity grade standard of WHO, the scoring system showed the sensitivity of 70.25%, the specificity of 85.53%, the false negative rate of 29.75%, the false positive rate of 14.47%, the positive predictive value of 71.43%, and the negative predictive value of 84.81%, with coincidence rate 0.80, while its area under the ROC was 0.765 (SE=0.027; 95% CI = 0.713-0.818; P< 0.001).” in pages 9-10, lines 626-713.
Comment 15 [Conclusions Section: Restructure conclusions based on the objectives of the study]
Response 15: Thank you for pointing this out. We restructured the conclusions based on the objectives of the study “In conclusion, a simplified scoring system with points ranging 0-11 was developed, and has been confirmed with high prediction value in predicting the risk of severe RSV-ALRI in children aged 0-16 years. Moreover, some prospective research work in multi-centers should be carried out to test its universality in clinical practice” This change can be found in page 11, lines 757-760.
Reviewer 2 Report
Comments and Suggestions for Authors
In this study, the author developed a scoring system to determine the severity of RSV in children. The authors utilized some factors such as age below 2, siblings, CHD, ..etc to build up a system of point up to 11.
The authors assessed that the risk of 5 points is the cut off where point abiove them represent severe and point below it is moderate,
Major points:
1) THe rationale of chossing these factors is not clear.
2) How did the authors calculate OR value from which they determined how many point to each factor
3) What about mild disease?
4) Why clinical symptoms are not included in the score?
5) What about vaccination of RSV and score? also respiratory track infection
Comments on the Quality of English LanguageModerate Language editing
Author Response
Comment 1) The rationale of chossing these factors is not clear.
Response 1: Thank you for pointing this out. We explained the rationale of choosing these factors as “According to the systematic review and meta-analysis, it has been reported that there were several factors associated with RSV-sALRI in young children (younger than 2 years), prematurity, low birth weight, being male, having siblings, maternal smoking, history of atopy, no breastfeeding, crowding and congenital heart disease [13, 14]. It also has been reported that there are two major RSV subtypes, A and B, and multiple genotypes, which can coexist during RSV epidemic season every year and result in different disease severity [15] ” in page 2, lines 16-21. “However, there are some limitations in the study. Firstly, many risk factors to RSV-sALRI reported in previous studies [31, 32], such as race/ethnicity, parents’ educational level or social conditions, allergy history of family, and exposure to cigarette smoking, were not evaluated in the study for only one race/ethnicity and the lack of information about parents’ educational level or social conditions, allergy history of family, exposure to cigarette smoking” in page 11, lines 712-716.
Comment 2) How did the authors calculate OR value from which they determined how many point to each factor
Response 2: Thank you for pointing this out. We added “The OR value was calculated by dividing the ratio of positive to negative numbers in the case group (severe RSV–associated ALRI group) from the ratio of positive to negative numbers in the control group (moderate RSV-associated ALRI group)” in page 3, lines 371-373, and “The best performance model was used to construct the scoring system, in which rounding-off method (decimals to round up and round down number) was chosen to convert the ORs into score points” in page 4, lines 429-431.
Comment 3) What about mild disease?
Response 3: Thank you for pointing this out. We defined the severity of disease as “According to the World Health Organization (WHO), the severity of ALRI is classified into three grades: â‘ mild-ALRI, with acute lower respiratory tract infection symptoms +, not requiring hospitalization; â‘¡ moderate-ALRI, with acute lower respiratory tract infection symptoms +, requiring hospitalization, blood oxygen saturation ≥93%; â‘¢ severe-ALRI, with acute lower respiratory tract infection symptoms +, requiring hospitalization, and blood oxygen saturation <93% [19]. In this study, only hospitalized patients with moderate- and severe-ALRI were included.” This change can be found in page 2, lines 188-193.
Comment 4) Why clinical symptoms are not included in the score?
Response 4: Thank you for pointing this out. We only evaluated disease severity by “According to the World Health Organization (WHO), the severity of ALRI is classified into three grades: â‘ mild-ALRI, with acute lower respiratory tract infection symptoms +, not requiring hospitalization; â‘¡ moderate-ALRI, with acute lower respiratory tract infection symptoms +, requiring hospitalization, blood oxygen saturation ≥93%; â‘¢ severe-ALRI, with acute lower respiratory tract infection symptoms +, requiring hospitalization, and blood oxygen saturation <93% [19]. In this study, only hospitalized patients with moderate- and severe-ALRI were included.” This change can be found in page 2, lines 188-193.
Comment 5) What about vaccination of RSV and score? also respiratory track infection
Response 5: Thank you for pointing this out. It’s a good question. More data should be accumulated to track the severity of RSV-ALRI following the use of RSV vaccine. However, the clinical data were collected during 2009 to 2021, and no RSV vaccination was approved.
Reviewer 3 Report
Comments and Suggestions for Authors
Dear authors,
I have now completed the review of the manuscript titled "A scoring system to predict severe acute lower respiratory infection in children caused by respiratory syncytial virus."
In the present study, you present a promising scoring approach to predict RSV-sALRI in children. The manuscript is interesting and, in general, fairly well-written. However, I have some suggestions to further improve the quality of the manuscript. I would like to suggest that you address these limitations in the article, either by discussing them in the limitations section or, where feasible, by making the appropriate revisions:
1. The incidence of healthcare utilization for respiratory infectious diseases in children before and during the COVID-19 pandemic has changed. Can you discuss how it will affect the current results and whether your ideas are still valid in 2024?
2. The study only included patients admitted to the ICU and respiratory medicine department, lacking representation of mild ALRI cases. This limits the ability to distinguish risk factors between mild, moderate, and severe disease. How may the research be generalized to mild ALRIs?
3. While the simplified scoring system enables easy assessment of RSV-sALRI risk, how it would be practically implemented in a clinical setting is not fully addressed. It seems like some authors are working in a children's hospital, so providing guidance on when and how to use the scores and defining risk categories could enhance clinical utility.
Thank you for your valuable contributions to our field of research. I look forward to receiving the revised manuscript.
Author Response
Comment 1. The incidence of healthcare utilization for respiratory infectious diseases in children before and during the COVID-19 pandemic has changed. Can you discuss how it will affect the current results and whether your ideas are still valid in 2024?
Response 1: Thank you for pointing this out. It’s a good question. Previous research reports in our laboratory have shown that NPIs launched in Beijing seriously interfered with the endemic season of RSV, and the percentage of severe pneumonia patients decreased to 40.51% after NPIs launched. [Jiang ML, et al. Changes in endemic patterns of respiratory syncytial virus infection in pediatric patients under the pressure of nonpharmaceutical interventions for COVID-19 in Beijing, China. J Med Virol. 2023 ;95(1):e28411.]. In our opinion, NPIs launched in Beijing had no effect on the development of the scoring system in the study focusing on the risk factors. However, “Moreover, some prospective research work in multi-centers should be carried out to test its universality in clinical practice.” to answer the question whether our ideas are still valid in 2024,
Comment 2. The study only included patients admitted to the ICU and respiratory medicine department, lacking representation of mild ALRI cases. This limits the ability to distinguish risk factors between mild, moderate, and severe disease. How may the research be generalized to mild ALRIs?
Response 2: Thank you for pointing this out. It’ a pity that mild-ALRI outpatients were not included in the retrospective study. Therefore, we concluded that “Moreover, some prospective research work in multi-centers should be carried out to test its universality in clinical practice.” In the prospective study, these mild-, moderate-, and severe-ALRI groups will be included.
Comment 3. While the simplified scoring system enables easy assessment of RSV-sALRI risk, how it would be practically implemented in a clinical setting is not fully addressed. It seems like some authors are working in a children's hospital, so providing guidance on when and how to use the scores and defining risk categories could enhance clinical utility.
Response 3: Thank you for pointing this out. On the basis of the study, we will apply for an enrolled academic committee meeting to discuss the guidance on when and how to use the scoring system and defining risk categories which could enhance clinical utility.
Round 2
Reviewer 1 Report
Comments and Suggestions for Authors
I have no further comments or suggestions about the manuscript. The authors have responded to my previous comments to the best of their ability.
Comments on the Quality of English LanguageMinor editing of English language required
Reviewer 2 Report
Comments and Suggestions for Authors
No furthers comments
Comments on the Quality of English Languagemoderate editing
Reviewer 3 Report
Comments and Suggestions for Authors
All comments were addressed.